# Volumetric Flow Assessment in Doppler Ultrasonography in Risk Stratification of Patients with Internal Carotid Stenosis and Occlusion

**DOI:** 10.3390/jcm11030531

**Published:** 2022-01-20

**Authors:** Piotr Kaszczewski, Michał Elwertowski, Jerzy Leszczyński, Tomasz Ostrowski, Zbigniew Gałązka

**Affiliations:** Department of General, Endocrine and Vascular Surgery, Medical University of Warsaw, Banacha 1A, 02-091 Warsaw, Poland; elwertowski.michal@gmail.com (M.E.); tostr@vp.pl (T.O.); zbigniew.galazka@wum.edu.pl (Z.G.)

**Keywords:** Doppler ultrasonography, carotid stenosis, carotid artery disease, TIA, stroke, cerebral blood flow, cerebrovascular reserve

## Abstract

(1) Background: Alterations of blood flow volume in extracranial arteries may be related to the risk of occurrence of neurological symptoms. The aim of this study was the estimation of cerebral blood flow (CBF) in Doppler ultrasonography, as well as comparison of the flow volume in asymptomatic patients over 65 years old with ≥50%, and symptomatic patients with ≥70% internal carotid artery (ICA) stenosis, in order to assess whether the changes in the CBF correlates with the presence of neurological symptoms. (2) Methods: 308 patients over 65 years old were included in the retrospective cohort observational study: 154 asymptomatic with ≥50% ICA stenosis, 123 healthy volunteers, and 31 symptomatic referred for surgical treatment. The study group was split according to ICA stenosis (50–69%, 70–99% and occlusion). In all patients an extensive Doppler ultrasound examination with measurements of flow volume in common, internal, external carotid (ECA) and vertebral arteries (VA) was performed. (3) Results: Among asymptomatic (A) and symptomatic (S) patients with carotid stenosis 3 subgroups were identified: 57/154—37% (A) and 8/31—25.5% (S)—with significantly increased flow volume (CBF higher than reference range: average CBF + std. dev in the group of healthy volunteers), 67/154—43.5% (A) and 12/31—39% (S)—with similar to reference group flow volume (CBF within range average ± std.dev), and 30/154—19.5% (A) and 11/31—35.5% (S)—with decreased flow volume in extracranial arteries (flow lower than average-std.dev. in healthy volunteers). In symptomatic patients the percentage of patients with significant compensatory increased flow tends to raise with the severity of the stenosis, while simultaneous decline of number of patients with mild compensation (unchanged total CBF) is observed. The percentage of patients without compensation remains unchanged. In the group referred for surgical treatment (symptomatic, ≥70% ICA stenosis) the percentage of patients with flow compensation is twice as low as in the asymptomatic ones with similar degree of the ICA stenosis (8/31—25.8% vs. 26/53—49%, *p* = 0.04). Compensatory elevated flow was observed most frequently in ECA. (4) Conclusions: The presence of significant volumetric flow compensation has protective influence on developing ischaemic symptoms, including TIA or stroke. The assessment of cerebral inflow in Doppler ultrasonography may provide novel and easily accessible tool of identifying patients prone to cerebral ischaemia. The multivessel character of compensation with enhanced role of ECA justifies the importance of including this artery in the estimation of CBF.

## 1. Introduction

Nowadays computed tomography (CT), magnetic resonance imaging (MRI), B-mode + contrast enhanced ultrasound belongs to the standard of diagnosis of significant carotid artery stenosis; however, they do not answer the key clinical question: why do patients with similar narrowing of carotid vessels behave clinically so differently [1]? Therefore, we changed diagnostic approach from simple stenosis measurements towards comprehensive evaluation of global cerebral flow evaluation along with compensatory mechanisms, that enable undisrupted brain function despite severe stenosis or occlusion of one (or more) brain supplying vessels.

Cerebral blood flow (CBF) strongly correlates with cerebrovascular reserve (CVR), therefore with the risk of forthcoming ischaemic events, regardless of the presence of neurological symptoms, stenosis, or occlusion of the supplying artery, or cerebrovascular reactivity testing method [2].

The role of recruitable collateral circulation has recently become apparent as a reliable and sensitive predictor of occurrence of ischaemic symptoms (including stroke), their severity and clinical outcomes of the treatment or rehabilitation [3,4,5,6].

Cerebral haemodynamics is also a key factor influencing neurocognitive functioning in patients with severe ICA stenosis. The improvement of cognitive performance was observed after carotid endarterectomy (CEA) in patients with TIA and ipsilateral high-grade ICA, who initially had decreased values of CVR. The improvement correlated inversely with age and preoperative CVR values [7,8].

Indications for carotid revascularization include the presence of neurological symptoms in patients with significant >70% ICA stenosis; it reduces the risk of ipsilateral stroke, while in asymptomatic ones the benefit is less pronounced. The management of asymptomatic patients remains controversial [1,2,9,10,11].

The aim of this study was to propose a novel approach of stratifying risk of neurological incidents: a global cerebral inflow assessment in Doppler ultrasonography (DUS), which may change current policy and indications for surgical intervention. It is based on characterisation of the flow volume in extracranial vessels: ICA, ECA and VA in patients >65 years old, with ≥50% carotid stenosis. Additionally, to prove the clinical significance of compensatory mechanisms, we compared the flow between asymptomatic patients and the symptomatic ones, with ICA stenosis ≥70%, who were referred for surgical treatment in order to assess whether the flow volume changes correlate with the incidence of neurological symptoms.

## 2. Materials and Methods

This study included 308 patients. The group of 123 healthy volunteers without carotid stenosis served as the control group. The flow volume values, which in this study are hereafter called “reference” values or “proposed reference standard” were established by our team based on the abovementioned group and published in 2020 [12]—see Table 1 in the Section 3. In this study, the flow volumes in healthy volunteers were compared with the flow volume values in the patients with carotid artery disease.

The study group consisted of 185 patients divided into two subgroups.

The first one contained 154 asymptomatic (no neurological symptoms within preceding 6 months and in the previous medical history) patients over 65 years of age with ICA stenosis from 50% to occlusion (59 female; age 72.1 ± 6.3 years, 95 males; age 73.3 ± 6.9 years).

The second subgroup comprised 31 symptomatic (the presence of neurological symptoms within preceding 6 months) patients referred for surgical treatment (13 female; age 71.8 ± 5.6 years, 18 males; age 70.8 ± 7.1 years). Detailed data are presented in the Table 2. In order to determine clinical significance of the flow volume values, the flow in the asymptomatic subgroup was compared with the flow in symptomatic patients referred for carotid endarterectomy due to ≥70% ICA stenosis.

Recruitment of the patients to both groups in terms of concomitant disorders was conducted according to the previously described protocol, in order to eliminate its influence on cerebral blood flow volume [12]. The criterion of the presence carotid stenosis and presence of neurological symptoms was a prerequisite of selecting patients to asymptomatic group with ICA stenosis or the group referred for surgical treatment respectively.

Before the examination an informed consent had been given by all study participants. The study was held under the approval of the Medical University of Warsaw Bioethical Committee.

In all patients a DUS examinations measuring blood flow volume in internal carotid artery (ICA), external carotid artery (ECA) and vertebral artery (VA) were performed. Firstly, the diameter of each vessel was measured using three different techniques: B-mode, SMI (superb micro-vascular imaging) mode and B-mode combined with SMI image. The average of three measurements was considered a diameter of the vessel. Consecutively the flow volume in the artery was measured with the ultrasound scanner semiautomatic program (the flow volume was calculated based on the spectral Doppler and the diameter of the vessel)—see Figure 1. All measurements were carried out three times and their average was considered as the final result. The cerebral blood flow (CBF) was calculated as the grand total of the flow volumes in all aforementioned extracranial arteries—see Figure 2. Examinations were conducted following the previously described protocol [12], by the same experienced sonographer, using Canon Aplio i800 ultrasound scanner with Linear i11LX3 transducer (Canon Medical Systems Corporation, Otawara, Japan). The blood flow volumes were calculated using.

The ICA flow velocity measurements were conducted at the sites of maximal stenosis, as well as at the distal part of the artery (at least 3 cm distal to the stenotic area, where the flow regained its laminar character). The assessment of the severity of the stenosis was based on Results of Society of Radiologists in Ultrasound Consensus Conference on the diagnosis of Internal Carotid Artery Stenosis and DEGUM criteria—using the peak systolic velocity (PSV), end-diastolic velocity (EDV) changes as well as ICA PSV/CCA PSV ratio, post-stenotic flow disturbances (severity and length) and reduction. Additional criteria were used for ICA stenoses, that exceeds 80%: prolongation of acceleration time >0.4 s and flow velocity reduction in the upper portion of the vessel and the PSV ratio between stenosis and upper part over 8.0 [13,14,15,16].

Statistical analysis was performed with Statistica 13 (StatSoft Polska Sp. z o.o., Kraków, Poland).

For the comparison of the two groups the *t*-test and the Mann–Whitney U test were used. The Shapiro–Wilk test was performed as a test of normality. Levene’s test was used to assess the equality of variances. The normal distribution of data with equal variances was a prerequisite to use the *t*-test. With no equality of variances, the *t*-test with Cochran-Cox correction was performed. When one of the variables was of no normal distribution, the non-parametric Mann–Whitney U test was performed.

For multiple groups the sets of data in which all variables were of normal distribution were analysed with one-way analysis of variance. When the data were not following a Gaussian distribution, the non-parametric Kruskal–Wallis one-way analysis of variance test was performed. The statistical significance was established with post-hoc tests. In all tests the significance level was <0.05.

## 3. Results

### 3.1. CBF Reference Values

The reference values of the flow in extracranial arteries, based on the examination of the group of healthy volunteers, are presented in Table 1.

For the purpose of this study:

The values exceeding the proposed reference value: average + standard deviation are referred as to “compensatory increased flow” or “significant compensation.”

The values within proposed reference are referred to as “mild compensation” or ”similar to reference group flow volume”—in the presence of major reduction of flow in one of carotid arteries, the increase in the other vessels allows to maintain the CBF within proposed standards.

The blood flow volume, which was lower than proposed reference value: average—standard deviation, are referred as to “no compensation”.

### 3.2. CBF in Asymptomatic Patients with ≥50% ICA Stenosis

In the group of 154 asymptomatic patients with ICA stenosis ≥50%, the three subgroups were identified in comparison to the group of healthy volunteers:Patients with elevated CBF as a result of multivessel significant flow increase (57/154);Patients with less pronounced compensation (mild compensation) in other extracranial arteries, which results in CBF within proposed reference standard for healthy population (67/154);Patients with lower CBF than the proposed reference standards (30/154).

The percentage patients with flow compensation tend to increase with the severity of stenosis, and slightly decreases in the occlusion group. At the same time the percentage of patients without flow compensation does not significantly change and remain relatively constant, between 18.9% and 20%. The increase in percentage of people with compensatory elevated flow is accompanied by simultaneous decrease of patients with flow volume values “within proposed standard”.

#### 3.2.1. CBF in the Group with 50–69% ICA Stenosis

In patients with ICA stenosis between 50–69%, majority 34/66 (51.5%) presented the flow volume within the proposed values for the respective age group. A total of 13/66 (19.7%) had the flow below the reference values, while 19/66 (28.8%) presented increased CBF values.

#### 3.2.2. CBF in the Group with 70–99% ICA Stenosis

By way of contrast, in the group with ICA stenosis ≥70%, most frequently—that is in 26/53 (49%), a compensatory increased flow volume was observed. In 17/53 (32.1%) the flow remained within proposed reference values, while in 10 patients (18.9%) the flow was lower than proposed standard.

#### 3.2.3. CBF in the Group with ICA Occlusion

In the group with ICA occlusion 7/35 (20%) the flow was lower than proposed standard. In 16/35 patients (45.7%) the total CBF remained within the reference values. AN elevated blood flow volume values in extracranial arteries were observed in 12/35 patients (34.3%).

#### 3.2.4. CBF in Merged Group ICA 70–99% Stenosis + ICA Occlusion

If groups of patients with stenosis ≥70% and occlusion are merged, then: reduced flow volumes were observed in 17/88 (19.3%), mild compensation in 33/88 (37.5%) and elevated flow values in 43.2%—38/88 patients.

The differences in groups without flow compensation do not reach any significant level. The detailed data are presented in Table 3 and in the Figure 3.

### 3.3. Comparison of the Flow Volume Values and Flow Changes in Asymptomatic Patients ≥50% ICA Stenosis

The flow volume values of total cerebral inflow (mean, 25–75%, min–max), which were obtained in the group of asymptomatic patients with ≥50% ICA stenosis, are presented in Table 4 and in Figure 4.

In the groups of patients, aged 65–69, 70–74 and 75–79 years old significant flow differences were observed except for “reference” and “mild compensation”—see Figure 4A–C respectively.

In the group aged ≥80 years old, one patient with the flow of 555 mL/min presented no compensation and was excluded from the statistical analysis. Substantial differences were observed between: “significant compensation” and other subgroups—see Figure 4D.

### 3.4. Pathways of Volumetric Flow Compensation in the Group of Asymptomatic Patients with ≥50% ICA Stenosis

The volumetric flow compensation was observed most frequently in the ECA—in 238 arteries, less frequently in VA—150 arteries, and least frequently in the contralateral ICA—in 77 arteries.

### 3.5. CBF in Symptomatic Patients Referred for Surgical Treatment

Out of the group of 31 symptomatic patients referred for Carotid Endarterectomy 11/31 (35.5%) presented lower flow volume values than the proposed standard. In this subgroup, one major stroke and ten TIA were diagnosed.

A total of 12/31 (38%) had the flow within proposed norm for their age. In this group, four strokes and eight TIA were diagnosed.

In eight patients (24.5%) elevated cerebral blood flow volume values were observed. In this group one stroke and seven TIA were diagnosed. The data are presented in Table 6

Comparing asymptomatic and symptomatic patients with stenosis of 70–99% a statistically significant difference (*p* = 0.0413) in patients with flow compensation is observed: 26/53—49% vs. 8/31—25.8% in those referred for surgical treatment. The relative risk of observing the compensatory increased flow in extracranial arteries in asymptomatic patients (n = 53) is almost two times higher than in symptomatic group (n = 31)—RR = 1.9; *p* < 0.0413. Patients without flow compensation constituted 35.5% of referred for interventional procedure and only 19.3% (*p* = 0.08) of asymptomatic patients with more than 70% stenosis and occlusion group, and 18.9% (*p* = 0.06) of patients with ICA stenosis 70–99%.

### 3.6. CBF Comparison in Patients with ≥70% ICA Stenosis: Asymptomatic Group and Symptomatic, Referred for Surgical Treatment

In the subgroup with ICA stenosis ≥70% statistically lower flow volume was observed in symptomatic patients in comparison with asymptomatic ones.

The data concerning the differences in the flow volume between symptomatic and asymptomatic patients with ≥70% ICA stenosis are presented in the Figure 5 and Table 7.

The group aged 75–79 was excluded from the analysis (in the symptomatic group there were only two patients).

In age groups 65–69, 70–74 and >80 years old statistically significant differences in flow volume were observed.

## 4. Discussion

The main finding of our study is identifying that changes in CBF correlate with the incidence of ischemic symptoms in patients with ICA stenosis.

Among patients with carotid artery stenosis, both: asymptomatic and symptomatic the three subgroups with volumetric flow changes were identified: patients with elevated CBF as a result of multivessel significant flow increase, patients with mild, less pronounced, compensation in whom the CBF is within proposed reference standard for healthy population, and patients without compensation, with CBF lower than healthy, equally aged population.

The percentage patients with flow compensation tend to increase with the severity of stenosis, and slightly decreases in the occlusion group. At the same time, the percentage of patients without flow compensation does not significantly change and remain relatively constant, between 18.9% and 20%. The increase in percentage of people with compensatory elevated flow is accompanied by simultaneous decrease of patients with flow volume values “within proposed standard”.

What is prominent, in the group referred for surgical treatment (symptomatic, ≥70% ICA stenosis) the percentage of patients with flow compensation is twice as low as in the asymptomatic ones with similar degree of the ICA stenosis.

Aiming to highlight the idea of our study, the flow volumes between all age subgroups in patients with ICA stenosis equal to or more than 70% were compared. It shows that asymptomatic patients tend to have higher CBF than symptomatic groups. The shortcoming of this comparison is relatively small number of patients (the study design demands separate assessment of patients in different age groups 65–69, 70–74, 75–79 and ≥80 years old), therefore, despite statistically significant differences and clearly identifiable tendency we do not want to draw any conclusion concerning numbers and amount of flow compensation. Further studies on much larger groups of patients are required.

In our study, the decreased flow volume in extracranial arteries, the decreased CBF, is more frequently observed among symptomatic patients. While the tendency is clearly visible, the analysis of absolute numbers is very close to statistical significance, but due to relatively small number of patients does not reach it in Chi2 test (*p* = 0.06 and 0.08). If both groups would be two times larger, than *p* values would be below 0.05. It may indicate that asymptomatic patients with lower values of CBF—without volumetric flow compensation, may be featured with higher risk of occurrence of ischemic symptoms.

Atherosclerotic lesions in extracranial arteries are related to increased risk of ischemic symptoms, while surgical treatment leads to its reduction [17,18].

The incidence of stroke in symptomatic patients with significant carotid stenosis is up to 15% in one year following the diagnosis, which suggest that within this group there are patients more prone to developing severe ischaemic events [19].

A proper CBF allows for proper neuronal function. Gupta et al. have found distinct correlation between impairment in cerebrovascular reserve and increased risk of ischemic events in patients with carotid stenosis or occlusion. The authors suggested that CVR may be helpful in stroke risk stratification [2].

The strong correlation between impaired cerebrovascular reserve and the risk of forthcoming ischemic events was also reported by Reinhard et al. The authors stated that transcranial Doppler CO_2_ reactivity is useful in risk assessment in both: asymptomatic and symptomatic patients with carotid lesions [20].

In the study by Sobczyk et al., the authors using blood–oxygen-level-dependent MRI assessed regional cerebral blood flow changes in response to a hypercapnic stimulus. They emphasize that carotid artery stenosis cause impairment of cerebrovascular reactivity. The authors stressed (after finding patients with normal ipsilateral cerebrovascular reserve in groups with severe ICA stenosis as well as occlusion), that the degree of the development of collateral circulation cannot be predicted with the severity of the stenosis [3].

This finding can be explained only with the presence of well-developed and sufficient collateral circulation, which comprise of the intracranial and extracranial anastomoses. The former includes vertebrobasilar circulation, the circle of Willis (CoW), tectal plexus, anterior, middle and posterior cerebral arteries branches, as well as leptomeningeal anastomoses. The latter covers orbital plexus and rete mirabile caroticum connecting ICA and ECA vascular beds. It appears that the collateral circulation plays a key role in determining the localization, type, and the extent of stroke [21,22,23].

In our study, in the group of symptomatic patients with carotid artery stenosis, the vast majority have CBF within or below the reference values. It is noteworthy that in symptomatic patients referred for interventional treatment a significant, nearly two-fold, increase of percentage of patients without flow compensation was observed (18.9% vs. 35.5%). What is also apparent, two-fold difference in percentage of patients with significant flow compensation was observed in asymptomatic patients comparing to those referred for surgical treatment (49% vs. 24.5%). Similar relations were observed if a merged groups of asymptomatic patients: those with stenosis 70–99% and occlusion are compared with symptomatic ones. This clearly shows that volumetric flow assessment in Doppler ultrasonography can, and should be used in risk stratification of asymptomatic patients with ICA stenosis.

Generally, CoW is regarded as the most important collateral pathway. However, Jongen et al. noted a gradual decrease in the cerebral blood flow in patients with increasing severity of carotid stenosis, which was independent of the CoW morphology, concluding that collateral pathways including ophthalmic and leptomeningeal vessels, may compensate for the CoW collaterals [24].

In this study, a particularly important role of external carotid artery in flow compensation was observed. In physiological conditions, ECA provides very little or no blood supply to the central nervous system; however, it is a vital collateral pathway in case of severe ICA stenosis. ECA endarterectomy in patients with ICA occlusion may benefit in reduction of neurological symptoms, such as amaurosis fugax, hemispheric TIA as well as non-lateralizing symptoms [25]. ECA blood flow may account for up to one-third of CBF in patients with bilateral ICA occlusion [26]. In patients with bilateral ICA stenosis the revascularization of significantly stenosed ECA may result in significant (15–39% ipsilateral and 12–52% contralateral) cerebral blood flow volume increase [27].

It has recently become apparent that the presence of collateral circulation, which is recruitable, is a powerful predictor of occurrence of ischaemic symptoms (including stroke), their severity and clinical outcomes of either treatment or rehabilitation [3,4,5,6].

Lattanzi et al. observed that the increase in the CVR following carotid endarterectomy in patients with high grade ICA stenosis is related to improvement in postoperative cognitive function. The authors examined the group of patients with history of transient ischemic attack within the past 6 months and ipsilateral high-grade ICA stenosis. With the examination of the cerebral vasomotor reactivity to hypercapnia, measured through transcranial Doppler ultrasonography authors assessed cerebral hemodynamics. In the two studies, authors used tests assessing right and left hemisphere cognitive functions (coloured progressive matrices, complex figure copy test, phonemic plus categorical verbal fluency tests). They showed that the postoperative improvement in cerebral vasomotor reactivity is connected with better neurocognitive function. The inverse association of cognitive improvement with the age of the patient was also observed [7,8].

Our method may also be useful in identifying patients, who may potentially benefit from CEA in terms of postoperative improvement in neurocognitive functions.

According to some authors the global cerebral inflow assessment with Doppler ultrasonography as well as other imaging methods measuring resting blood flow, are not able to deliver information concerning recruitable perfusion [3,28].

The development of collateral circulation begins immediately after the flow pressure drops, as a result of myogenic regulation, which is most prominent in small cortical arterioles [17,29].

The early phase of collaterals formation is a result of arteriogenesis—development of functional collateral vessels from pre-existing arterioles, induced mainly by mechanical factors. Angiogenesis, the formation of new vessels as a response to angiogenic factors, may also play a role in the development of collateral circulation; however, it is a much longer process [30,31].

In our study, the patients with relatively long medical history of either stenosis or occlusion were included. In such patients all mechanisms of the recruitment of collateral circulation had enough time to develop. Therefore, increased flow volume in extracranial arteries may be observed in patients with high cerebrovascular reactivity and reserve, while patients with less pronounced compensation may have worse collateral circulation and lower cerebrovascular reserve. The assessment of cerebral inflow in Doppler ultrasonography indeed does not directly show the potential to recruit collateral circulation, however, the significant increase in global cerebral inflow can occur only in patients with such collaterals being well developed.

Comparing flow volume values to those in healthy volunteers, especially in individual arteries, using sonographic methods, can indirectly assess both the potential of the collateral circulation to form, and the degree of its development. The major advantage of our method is its availability. While nuclear medicine methods, MRI, positron emission tomography–computed tomography (PET-CT), single-photon emission computed tomography (SPECT) or transcranial Doppler are mainly available in specialized vascular centers, Doppler ultrasonography may be performed even in an outpatient clinic.

A novel tool in the diagnostic of carotid artery disease is a three-dimensional ultrasound. Contemporarily it is mainly used for carotid plaque assessment: its morphology, the presence of ulceration, which are correlated with risk of developing ischemic symptoms. It is featured with high interobserver and interobserver reproducibility and can identify plaque volume changes as low as 4–6% with 95% confidence [32,33,34].

It has been recently proven that three-dimensional ultrasound may be a novel, promising, potentially easily accessible and accurate tool in quantification of volumetric blood flow [35].

The presence of collateral circulation promotes patient survival. It was proved that in patients with well-developed collateral circulation, who receive best medical therapy, the risk of ischemic stroke was lower than in the group without collateralization of cerebral circulation (13.3% versus 6.3%, for disabling or fatal stroke; 27.8% versus 11.3% for hemispheric stroke) [36].

Our method may allow to identify asymptomatic patients, with significant ICA stenosis and diminished CBF. Such patients may be more prone to developing ischemic and perhaps may benefit from surgical intervention. However, such conclusions cannot be drawn based on current work. A prospective study on larger groups of patients should be conducted to investigate this issue.

## 5. Conclusions

The significant compensatory elevation of CBF was more frequently observed in asymptomatic patients, which suggest its protective influence on developing ischaemic symptoms.

The assessment of global cerebral inflow in Doppler ultrasonography may provide novel and easily accessible tool of identifying patients more prone to cerebral ischaemia—further studies on larger groups of patients are required.

The multivessel character of compensation with enhanced role of ECA stresses the importance of including this artery in the estimation of CBF.

## Figures and Tables

**Figure 1 jcm-11-00531-f001:**
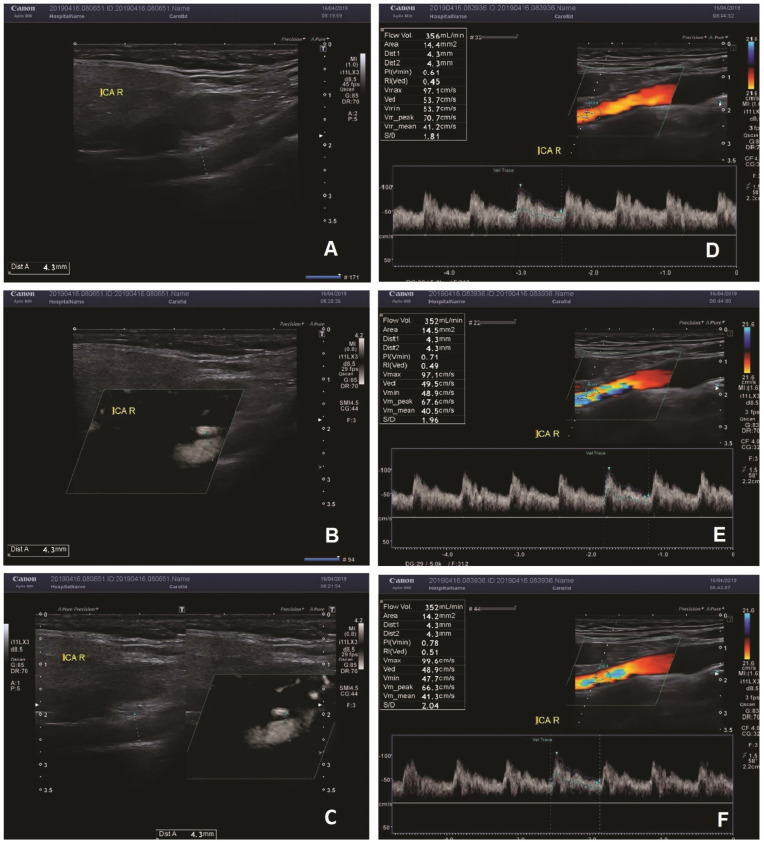
(**A**–**C**)—the measurement of the diameter of the vessel with B-mode (**A**), SMI (**B**), combined B-mode/SMI (**C**). In all three measurements, the diameter of the vessel was 4.3 mm. (**D**–**F**)—measurement of the flow volume in the right ICA. The flow volume was 356 mL/min in one measurement (**A**), and 352 mL/min in two measurements (**E**,**F**).

**Figure 2 jcm-11-00531-f002:**
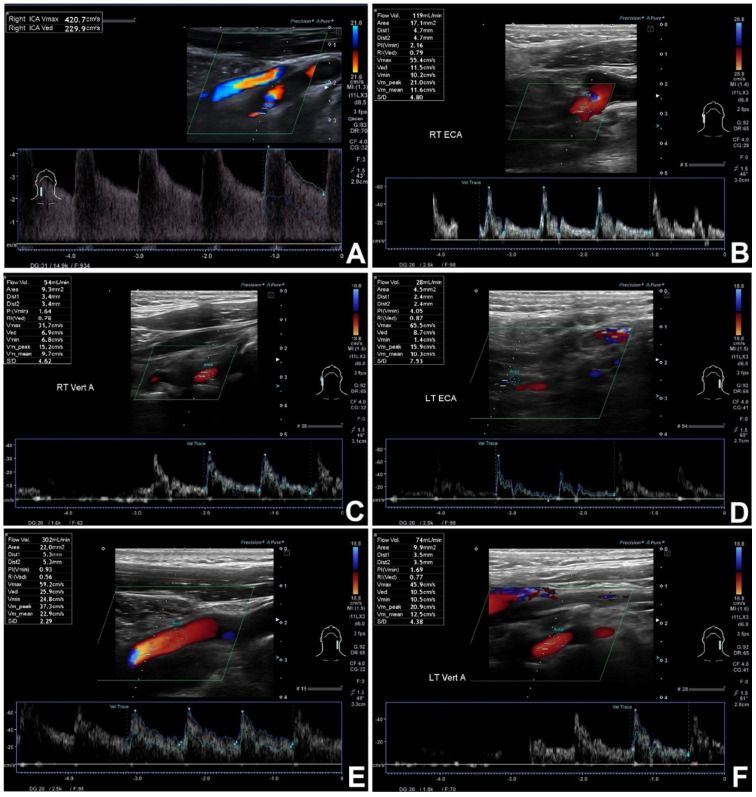
CBF estimation in symptomatic 67 years old patients with significant RICA stenosis and no volumetric flow compensation in other extracranial arteries. (**A**)—RICA—significant >95% stenosis, flow volume distally 43 mL/min. (**B**)—RECA with flow volume of 119 mL/min, (**C**)—RVA with flow volume of 54 mL/min, (**D**)—LECA with flow volume of 28 mL/min, (**E**)—LICA with flow volume of 302 mL/min, (**F**)—LVA with flow volume of 74 mL/min. CBF of 620 mL/min—no compensation.

**Figure 3 jcm-11-00531-f003:**
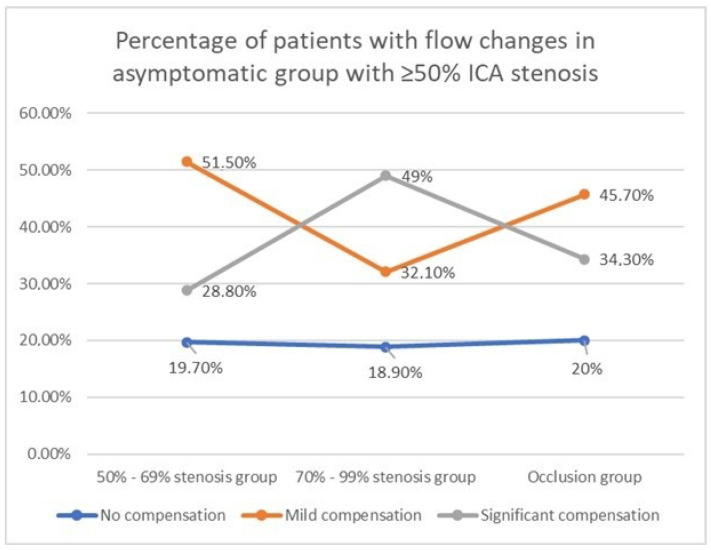
Differences in the percentage of patients with the flow volume changes in the asymptomatic group.

**Figure 4 jcm-11-00531-f004:**
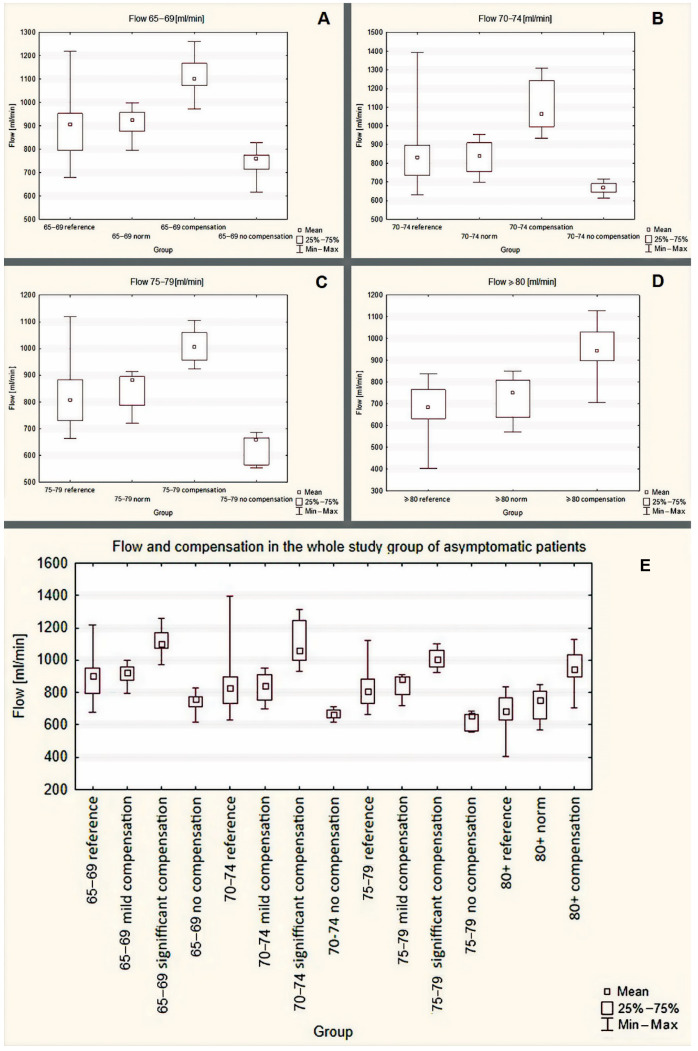
CBF volume values in the asymptomatic patients with internal carotid artery stenosis and occlusion (merged subgroups with different degree of ICA stenosis and occlusion). (**A**): 65–69 years old, (**B**): 70–74 years old, (**C**): 75–79 years old, (**D**): ≥80 years old, (**E**)—whole group merged together. *p* values are presented in the Table 5.

**Figure 5 jcm-11-00531-f005:**
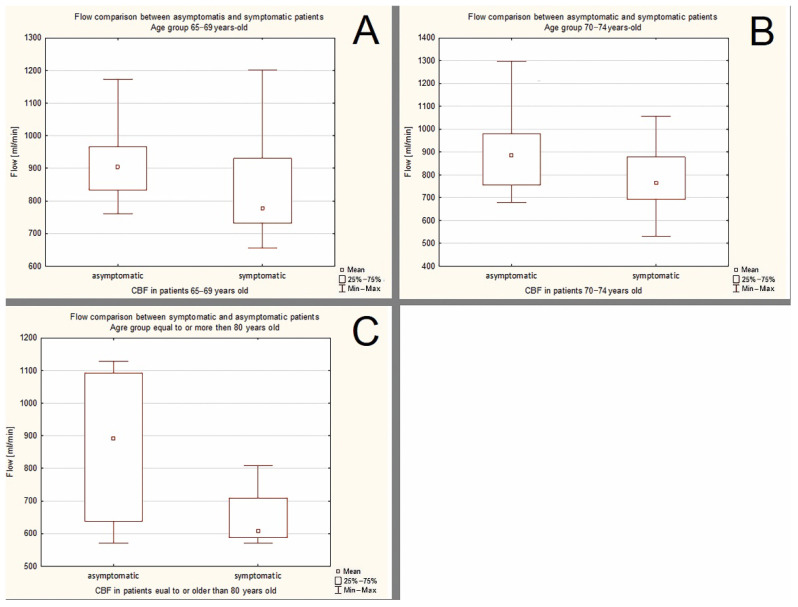
The differences in the flow volume between symptomatic and asymptomatic patients with ≥70% ICA stenosis. (**A**)—age group 65–69 years old. (**B**)—age group 70–74 years old. (**C**)—age groups ≥80 years old.

**Table 1 jcm-11-00531-t001:** Reference flow volume values in extracranial arteries RICA, LICA—right and left internal carotid artery. RECA, LECA—right ang left external carotid artery. RVA, LVA—right and left vertebral artery. Values are presented as average—standard deviation. Adapted from: Kaszczewski, P.; Elwertowski, M.; Leszczynski, J.; Ostrowski, T.; Galazka, Z. Volumetric Carotid Flow Characteristics in Doppler ultrasonography in Healthy Population Over 65 Years Old. *J. Clin. Med.* **2020**, *9*, 1375 [12].

Group—Age	65–69	70–74	75–80	>80
Proposed reference value (mL/min)	898.5 ± 119.1	838.5 ± 148.9	805.1 ± 99.3	685.7 ± 112.3
RICA (mL/min)	271.1 ± 63.6	236.0 ± 66.1	234.8 ± 62.3	202.3 ± 38.4
RECA (mL/min)	106.1 ± 35.0	103.7 ± 33.2	94.0 ± 24.14	83.1 ± 36.3
RVA (mL/min)	58.7 ± 29.1	60.2 ± 26.7	62.3 ± 28.4	55.7 ± 24.1
LICA (mL/min)	276.4 ± 57.5	239.8 ± 42.4	245.5 ± 32.3	204.4 ± 47.0
LECA (mL/min)	101.4 ± 30.9	104.7 ± 32.5	89.0 ± 21.9	79.0 ± 33.7
LVA (mL/min)	84.9 ± 33.0	80.4 ± 29.8	70.0 ± 21.5	58.8 ± 13.0

**Table 2 jcm-11-00531-t002:** Characteristics of the study group: 154 asymptomatic patients with equal or >50% ICA stenosis and 31 symptomatic patients referred for surgical treatment.

Study Group—Asymptomatic Patients with ICA 50% Stenosis—Occlusion			
ICA Stenosis (%)	50–69%	70–99%	100%—occlusion of the ICA
Number of patients	66	53	35
Average age (years)	72.2	73.4	71.8
Number of females	31	17	11
Number of males	35	36	24
Study group—symptomatic patients referred for CEA due to ≥70% ICA stenosis			
Average age (years)	71.8		
Number of females	13		
Number of males	18		

**Table 3 jcm-11-00531-t003:** Characteristics of the flow in asymptomatic patients with carotid stenosis.

	No Compensation	Mild Compensation Resulting in Flow within Proposed Standard for Age	Significant Compensation—Increased Flow Volume
	(Number of Patients)	(Number of Patients)	(Number of Patients)
% of people in 50–69% stenosis group	13/66—19.7%	34/66—51.5%	19/66—28.8%
% of people in 70–99% stenosis group	10/53—18.9%	17/53—32.1%	26/53—49%
% of people in occlusion group	7/35—20%	16/35—45.7%	12/35—34.3%
% of people in stenosis 70–99% + occlusions group	17/88—19.3%	33/88—37.5%	38/88—43.2%

**Table 4 jcm-11-00531-t004:** Total CBF in the group of asymptomatic patients with carotid stenosis.

	Flow Volume (mL/min) Mean 25–75% Min–Max			
Age Group (Years)	Reference (Healthy Volunteers)	No Compensation	Mild Compensation	Significant Compensation
65–69	905.5	758	923	1102,5
	796–953	714.5–775.5	876–958	1073–1167.5
	679–1220	616–828	796–999	973–1262
70–74	830	667.5	839	1064
	735–896	645–691	755–909	996–1243
	630–1394	614–715	697–953	934–1310
75–79	807	650	881	1005
	730.5–883.5	564–665	787–895	956.5–1060
	662–1119	552–685	720–913	923–1105
≥80	684.5	1 patient with flow of 555 mL/min (excluded from analysis)	751	943
	631.5–766		638–809	899–1032
	404–838		571–851	706–1129

**Table 5 jcm-11-00531-t005:** Flow differences between the groups—*p* values.

	65–69 Reference	65–69 Mild Compensation	65–69 Signifficant Compensation	65–69 No Compensation	70–74 Reference	70–74 Mild Compensation	70–74 Signifficant Compensation	70–74 No Compensation	75–79 Reference	75–79 Mild Compensation	75–79 Signifficant Compensation	75–79 No Compensation	80+ Reference	80+ Mild Compensation	80+ Signifficant Compensation
65–69 reference		0.9	0.002	0.000	0.008	0.025	0.000	0.000	0.002	0.100	0.007	0.000	0.000	0.000	0.62
65–69 mild compensation			0.000	0.000	0.000	0.300	0.000	0.000	0.003	0.007	0.001	0.000	0.000	0.000	0.128
65–69 signifficant compensation				0.000	0.000	0.000	0.436	0.000	0.000	0.000	0.008	0.000	0.000	0.000	0.000
65–69 no compensation					0.018	0.019	0.000	0.004	0.040	0.004	0.000	0.004	0.125	0.730	0.000
70–74 reference						0.960	0.000	0.001	0.987	0.930	0.000	0.000	0.000	0.014	0.002
70–74 mild compensation							0.002	0.002	0.980	0.930	0.000	0.000	0.000	0.014	0.002
70–74 signifficant compensation								0.000	0.000	0.000	0.130	0.001	0.000	0.000	0.012
70–74 no compensation									0.000	0.000	0.000	0.300	0.900	0.300	0.000
75–79 reference										0.980	0.006	0.022	0.001	0.041	0.002
75–79 mild compensation											0.120	0.010	0.000	0.003	0.006
75–79 signifficant compensation												0.000	0.000	0.000	0.241
75–79 no compensation													0.184	0.119	0.000
80+ reference														1.000	0.000
80+ mild compensation															0.000

**Table 6 jcm-11-00531-t006:** Characteristics of the flow and clinical features of the group of symptomatic patients referred for surgical treatment.

	No Compensation (Number of Patients)	Mild Compensation Resulting in Flow within Proposed Standard for Age (Number of Patients)	Significant Compensation Resulting in Increased Flow Volume (Number of Patients)
Symptomatic patients with significant ICA stenosis	11	12	8
% of the group	11/31—35.5%	12/31—39%	8/31—25.5%
Number of strokes	1	4	1
Number of TIA	10	8	7

**Table 7 jcm-11-00531-t007:** Detailed data concerning differences in the flow volume between symptomatic and asymptomatic patients with ≥70% ICA stenosis.

Age Group	Clinical Presentation	Mean (mL/min)	25–75% (mL/min)	Min–Max (mL/min)	*p* Value
65–69	Asymptomatic	902.5	833.5–965.5	761–1172	*p* = 0.049
	Symptomatic	777.5	732–930.5	656–1201	
70–74	Asymptomatic	884	756.5–981	679–1298	*p* = 0.041
	Symptomatic	763	692–878	530–1057	
≥80	Asymptomatic	891.5	638–1092	571–1129	*p* = 0.044
	Symptomatic	607	587–709.5	570–809	

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
