# Peer review of "Volumetric Flow Assessment in Doppler Ultrasonography in Risk Stratification of Patients with Internal Carotid Stenosis and Occlusion"

_jcm, 2022, doi:10.3390/jcm11030531_

Round 1

Reviewer 1 Report

This study aimed to characterize the flow volume in extracranial vessels in patients with internal carotid stenosis and occlusion.

The paper is overall interesting. There are, however, some issues that need to be further addressed.

It would be fine to expand a little bit the evidence about the impairment in cerebrovascular reserve as a determinant of the consequences of carotid stenosis, including not only cerebral ischemia but also cognitive dysfunction, and how the impairment in cerebral hemodynamics may be reversible after stenosis correction and it may help to identify those patients who may benefit mostly from stenosis correction (Ref. Neurocognitive functioning and cerebrovascular reactivity after carotid endarterectomy. Neurology 2018; Predictors of cognitive functioning after carotid revascularization. J Neurol Sci 2019).

Author Response

Dir Sir or Madam,

On behalf of all authors, we would like to thank you for time and effort devoted to preparing the review of our article. We appreciate Your suggestions concerning expanding the manuscript content with evidences about the influence of the surgical treatment on neurocognitive performance of the patients. That remark is really important to us, it may provide a new insight into our research and allow us to broaden our investigation in the field of cognitive function of patients following ICA stenosis treatment.

Following the review we implemented several amendments into our manuscript.  All the changes are marked with red color in order to facilitate the review process. The detailed list of the changes in the manuscript cover:

  1. The paragraph: “Cerebral haemodynamics is also a key factor influencing neurocognitive functioning in patients with severe ICA stenosis. The improvement of cognitive performance was observed after carotid endarterectomy (CEA) in patients with TIA and ipsilateral high-grade ICA, who initially had decreased values of CVR. The improvement correlated inversely with age and preoperative CVR values (7,8)” was added to the introduction part. New references were added:

Lattanzi, S., Carbonari, L., Pagliariccio, G., Bartolini, M., Cagnetti, C., Viticchi, G., Buratti, L., Provinciali, L., & Silvestrini, M.). Neurocognitive functioning and cerebrovascular reactivity after carotid endarterectomy. Neurology, 2018, 90(4), e307–e315 

Lattanzi, S., Carbonari, L., Pagliariccio, G., Cagnetti, C., Luzzi, S., Bartolini, M., Buratti, L., Provinciali, L., & Silvestrini, M. Predictors of cognitive functioning after carotid revascularization. J Neurol Sci. 2019; 405:116435.

  1. Following the suggestion of the Reviever 2 the definition of symptomatic and asymptomatic patients was precised: asymptomatic (no neurological symptoms within preceding 6 moths and in the previous medical history) symptomatic (the presence of neurological symptoms within preceding 6 months) patients. The criteria of patients selection together with detailed study protocol described in our previous study: Kaszczewski, P.; Elwertowski, M.; Leszczynski, J.; Ostrowski, T.; Galazka, Z. Volumetric Carotid Flow Characteristics in Doppler Ultrasonography in Healthy Population Over 65 Years Old. J. Clin. Med. 2020, 9, 1375. https://doi.org/10.3390/jcm9051375.
  2. In the discussion the paragraphs concerning improvement of neurocognitive functioning of the patients after ICA stenosis treatment have been added: Lattanzi et. al observed that the increase in the CVR following carotid endarterectomy in patients with high grade ICA stenosis is related to improvement in postoperative cognitive function. The authors examined the group of patients with history of transient ischemic attack within the past 6 months and ipsilateral high-grade ICA stenosis. With the examination of the cerebral vasomotor reactivity to hypercapnia, measured through transcranial Doppler ultrasonography authors assessed cerebral hemodynamics. In the two studies, authors used tests assessing right and left hemisphere cognitive functions (Coloured Progressive Matrices, Complex Figure Copy Test, phonemic plus categorical Verbal Fluency tests). They showed that the postoperative improvement in cerebral vasomotor reactivity is connected with better neurocognitive function. The inverse as-sociation of cognitive improvement with the age of the patient was also observed [7,8].

Our method may also be useful in identifying patients, who may potentially benefit from CEA in terms of postoperative improvement in neurocognitive functions.

  1. The spelling and grammatic mistakes have been corrected – all the changes are marker with red color.
  2. In the references: the reference number 3 was doubled. The references were updated and corrected.

We do hope that you will find the amended version of our manuscript interesting and suitable for publication in the Journal of Clinical Medicine.

Faithfully Yours,

Piotr Kaszczewski

Michał Elwertowski

Jerzy Leszczyński

Tomasz Ostrowski

Zbigniew Gałązka

Reviewer 2 Report

The authors investigated the usefulness of doppler ultrasonography (DUS) on the volumetric flow assessment in patients with carotid stenosis. The authors recruited patients with asymptomatic carotid stenosis (>=50% ICA stenosis) and symptomatic carotid stenosis (>=70% ICA stenosis referred for operation). Volumetric flow amounts were compared with a previously-established "reference value" in healthy adults by the same group. The authors found that the percentage of patients with compensatory increased flow would increase along with the degree of stenosis, until total occlusion. They also found that the majority of compensatory flow came from ECA.

With such an ambitious intention to quantify the compensatory flow in ICA stenosis, though, the readers may find the study was poorly organized and presented. The main concerns include:

1) The definition of "asymptomatic" and "symptomatic" is vague. How are asymptomatic group recruit? What are the indication for symptomatic patients?

2) No vascular risk factors in each group were presented, except basic age and sex.

3) Categorized into 50-70%, 70-99%, and total occlusion will loss the statistical power. The authors may find linear association between degree of stenosis and flow. In such way, maybe a classical reversed-V shape associations may be found between the degree of stenosis and compensatory flow.

4) No any description regarding the actual compensatory flow in ECA, VA and contralateral ICA. Not to say the differences between two groups.

5) The statistical analyses done are way too simple and redundant. No regression was done to identify independent association between degree of stenosis or symptomatic/asymptomatic and flow amount. 

6) The discussion is way too length and irrelevant to the present study. For example the whole circle of Willis and cerebrovascular reactivity parts were not the major findings in the results. 

In conclusion, this manuscript is too preliminary in its form. It should be radically reorganized and reanalyzed before submission.

Author Response

Dir Sir or Madam,

On behalf of all authors, we would like to thank you for the time and effort devoted to preparing a review of our article. We would like to thank you for the remarks to the manuscript that, to our surprise and joy, are consistent with further directions of our research. Therefore, we do hope to clarify your doubts, and prove the clinical importance of our research.

As the introduction to our response, we would like to present the idea of our research. We wanted to merge the current ultrasound and clinical diagnostic approach of ICA stenosis, based on flow velocity changes and the presence of neurological symptoms within preceding 6 months (which nowadays are mainly assessed during qualification to surgical treatment) with volumetric flow measurements in extracranial arteries. We aim to prove that asymptomatic patients, who do not have flow compensation in other extracranial arteries are more prone to developing neurological symptoms. We hope that this approach may influence the indications to surgical treatment in asymptomatic patients, with significant ICA stenosis, and without flow volumetric compensation. Further investigation is on this topic is continued.

We are conscious that there are several factors increasing the risk of neurological symptoms such as: hypoechoic carotid plaque, microembolization in transcranial Doppler, rapid stenosis progression – which may indicate the need of early surgical treatment even in asymptomatic patients, or those with ICA stenosis between 50-70%. We hope that volumetric flow compensation may became one of such factors.

We do hope we will be able to clarify Your doubts concerning our work:

  1. Asymptomatic and symptomatic patients – according to the recommendations our team identified patients as symptomatic, when they presented neurological symptoms within preceding 6 months. As asymptomatic patients we understand those without any previous medical history of neurological symptoms within last six months. Additionally for the purpose of this study we selected patients without any previous history of neurological symptoms to theasymptomatic group. We hope we clarified this issue in our previous publication, however, if not in sufficient way, we added the following sentence to our study:

“asymptomatic (no neurological symptoms within preceding 6 months and previous medical history)” and “symptomatic (the presence of neurological symptoms within preceding 6 months) patients”

  1. The protocol of the recruitment of the patients in terms of vascular risk factors was thoroughly described in our previous publication, with detailed list of concomitant disorders, with detailed inclusion and exclusion criteria. In order not to make the manuscript excessively lengthy (the length of the discussion was also one of your remarks) we cited our previous work. If it is not sufficient, please provide us with clear recommendations what we have to add to our manuscript. I would like to add that the second reviewer of our study pointed out that the study design as well as inclusion and exclusion criteria are clear, coherent, and stringent.

  1. 4, and 5. Dir sir, please let me comment the remarks in one paragraph, because, I am certain, I may treat them as a whole. These remarks are indeed really important, and we completely agree with your opinion. Unfortunately, the analysis of the flow compensation in particular extracranial arteries was not the topic of this study. It demands completely different study design and a new cycle of articles. Our team is working on such articles. We are analyzing the flow compensation patterns in all extracranial arteries in patients with occlusion, stenosis 70-99% and stenosis 50-69%, both symptomatic and asymptomatic. Unfortunately, every single group in terms of stenosis and age has to be analyzed separately. Performing a simple regression analysis has to be done on representative groups in order to have statistical power. In our study, the groups that may be compared are the patients with similar degree of stenosis: 70-99 % asymptomatic and symptomatic. We proved that the percentage of patients with and without compensation significantly differs. Our previous work showed that there were significant differences between age groups in terms of flow volume. Therefore to perform thorough statistical analysis in patients with different degree of stenosis and age groups the study group would be minimally about (40-50 patients x 4 age groups x 3 different stenosis degree x 2 – symptomatic/asymptomatic) 960 – 1200 patients. We are not able to gather homogenous study groups of that size in a single center study, or if yes, it will last at least 5-10 years.

Even in the analysis which we preform now (in the study about the patterns and the deerge of compensation), wanting to examine the degree and pattern of compensation, the amount of statistical analysis is extremely high, and the work has, I am afraid, statistical and mathematical character. We would like to present our results in one single work, however the very “Results“paragraph, will have about 15000 words, and in such work the main, clinical idea will be lost among mathematics, statistics, numbers and figures. Such work will be unbearable to read and to be analyzed. Therefore, we are preparing articles analyzing the flow in different stenosis groups separately. We hope You will have an opportunity to read or perhaps even review our future studies, that will, I do hope, answer all your questions and clarify all your doubts.

  1. I am afraid that the remarks to the discussion are a kind of opinion. There are no instructions concerning the modification and improvement of the work. In our opinion the discussion justifies and explains our idea, motivation and methods.

Dear reviewer, after reading the review we have the impression that you have a completely negative opinion about our article, its idea, and conclusions. We hope, we are wrong, and the remarks comes from misunderstanding of the idea of our research.

The main idea of this article is that with readily accessible tools (volume flow measurements are possible with practically every medium class ultrasound scanner) we propose NEW approach to serious clinical problem, which if applied can save life and life threating events in our daily routine work. We are clinicians: surgeons, diagnostic team members, not pure scientists, and we saw people dying in our ward and clinic because of stiff rules proposed by studies based on narrowing/symptomatology criteria. We apply the described method since 2015 starting with Aloka, and since 4 years Canon premium class scanner. Thank to this, we hope, we spared many of our patients from stroke and death – we feel clinical impact of this work is the most important factor that should be brought to everybody’s attention.

If You feel like we need further discussion, please write us an e-mail: [email protected] (MichaÅ‚ Elwertowski) or [email protected] (Piotr Kaszczewski).

Following the reviews, we implemented several amendments into our manuscript.  All the changes are marked with red color in order to facilitate the review process. The detailed list of the changes in the manuscript cover:

  1. The paragraph: “Cerebral haemodynamics is also a key factor influencing neurocognitive functioning in patients with severe ICA stenosis. The improvement of cognitive performance was observed after carotid endarterectomy (CEA) in patients with TIA and ipsilateral high-grade ICA, who initially had decreased values of CVR. The improvement correlated inversely with age and preoperative CVR values (7,8)” was added to the introduction part. New references were added:

Lattanzi, S., Carbonari, L., Pagliariccio, G., Bartolini, M., Cagnetti, C., Viticchi, G., Buratti, L., Provinciali, L., & Silvestrini, M.). Neurocognitive functioning and cerebrovascular reactivity after carotid endarterectomy. Neurology, 2018, 90(4), e307–e315 

Lattanzi, S., Carbonari, L., Pagliariccio, G., Cagnetti, C., Luzzi, S., Bartolini, M., Buratti, L., Provinciali, L., & Silvestrini, M. Predictors of cognitive functioning after carotid revascularization. J Neurol Sci. 2019; 405:116435.

  1. Following the suggestion of the Reviever 2 the definition of symptomatic and asymptomatic patients was precised: asymptomatic (no neurological symptoms within preceding 6 moths and in the previous medical history) symptomatic (the presence of neurological symptoms within preceding 6 months) patients. The criteria of patients selection together with detailed study protocol described in our previous study: Kaszczewski, P.; Elwertowski, M.; Leszczynski, J.; Ostrowski, T.; Galazka, Z. Volumetric Carotid Flow Characteristics in Doppler Ultrasonography in Healthy Population Over 65 Years Old. J. Clin. Med. 2020, 9, 1375. https://doi.org/10.3390/jcm9051375.
  2. In the discussion the paragraphs concerning improvement of neurocognitive functioning of the patients after ICA stenosis treatment have been added: Lattanzi et. al observed that the increase in the CVR following carotid endarterectomy in patients with high grade ICA stenosis is related to improvement in postoperative cognitive function. The authors examined the group of patients with history of transient ischemic attack within the past 6 months and ipsilateral high-grade ICA stenosis. With the examination of the cerebral vasomotor reactivity to hypercapnia, measured through transcranial Doppler ultrasonography authors assessed cerebral hemodynamics. In the two studies, authors used tests assessing right and left hemisphere cognitive functions (Coloured Progressive Matrices, Complex Figure Copy Test, phonemic plus categorical Verbal Fluency tests). They showed that the postoperative improvement in cerebral vasomotor reactivity is connected with better neurocognitive function. The inverse as-sociation of cognitive improvement with the age of the patient was also observed [7,8].

Our method may also be useful in identifying patients, who may potentially benefit from CEA in terms of postoperative improvement in neurocognitive functions.

  1. The spelling and grammatic mistakes have been corrected – all the changes are marker with red color.
  2. In the references: the reference number 3 was doubled. The references were updated and corrected.

We hope that our response will allow you to look at our study in a more favorable light and persuade you to give us a chance instead of rejecting the work.

Yours faithfully,

Piotr Kaszczewski,

Michał Elwertowski,

Jerzy Leszczyński,

Tomasz Ostrowski,

Zbigniew Gałązka

Reviewer 3 Report

dear authors,

this is an interesting study, but there are some issues that need clarification.

In the aim of the abstract please fill in that this is a comparison study and in the methods if this was a prospective observational study.

Did you add this in the clinicaltrialgov?

Page 1, line 20, 3 groups were identified.

Line 23, what is ‘compensatory’ increased flow.

Line 28-29: ‘has protective influence on developing ischaemic symptoms’ in the conclusion you should not provide outcomes on asymptomatic or symptomatic as you did not involve symptomatic patients in the comparison in the abstract. You should refer only on carotid disease as your control group is healthy volunteers.

You should not start you introduction so direct with the aim of the study, usually you should add this in the end of introduction.

Additionally in order to ’stratify risk of neurological incidents of patients’ you should have follow up with events (TIA or stroke) on this group. You do provide 31 patients who are symptomatic, but I had to read up to methods of the main text in order to find this out.

I would suggest to reform aim and abstract of the study. Currently it is not clear if your aim is to provide flow volume measurements for carotid stenosis or for symptoms.

Please better clarify your methodology of your study. Additionally, parts of the methods are outcomes and you should move them in results section.

You should add some images on blood flow measurement. You should add some ref for the 3 d ultrasound on carotid artery assessment.

Did you analyze the type of plaque with blood flow measurements? There could be a correlation between them.

Add subsections in the results area in order to be easier to understand the outcomes.

In the discussion area please start with your main findings and their importance and then you can elaborate on other studies.

Author Response

Dir Sir or Madam,

On behalf of all authors I would like to thank You for the time and effort devoted to prepare the review as well as for the constructive comments, which provided valuable insights, indispensable in refining manuscript content.

Having received the review, we put all our efforts to implement necessary changes and prepare the manuscript body in accordance with Your suggestions. We hope You will find the improved version of manuscript presenting aim of the study as well as results and discussion in more comprehensible manner. I would like to ensure Your Party that all  issues were thoroughly corrected, according to the review.

We would like to answer You questions before presenting the list of changes in the manuscript.

You asked whether our study has been registered in clinicaltrial.gov. No, this study was not registered in clinicaltrial.gov. We do not consider our study as a clinical trial because we did not change any algorithms of qualification of patients to surgical treatment. All symptomatic patients with ≥70% ICA stenosis were qualified for surgical treatment, according to current recommendations. All asymptomatic patients were on best medical treatment. However, we do hope to find a correlation between diminished CBF in asymptomatic patients and the risk of future ischaemic events in a prospective study. Perhaps our method may be in future a risk stratification tool, that may help to identify patients with high risk of occurrence of ischaemic symptoms and offer them early elective surgical treatment (like in case of the ulceration of the plaque, or microembolisation).

In order to facilitate the review process all the implemented changes were marked with a yellow colour.

Please find the detailed list of amendments below:

Introduction:

Background: the aim of the study was rewritten:

Lines 12-16: The aim of this study was the estimation of cerebral blood flow (CBF) in Doppler ultrasonography, as well as comparison of the flow volume in asymptomatic patients over 65 years old with ≥50%, and symptomatic patients with ≥70% internal carotid artery (ICA) stenosis, in order to assess whether the changes in the CBF correlates with the presence of neurological symptoms.

Matherials and methods:

Line 16: The “retrospective cohort observational study” was added. In our study we measured the flow volume in extracranial arteries. We afterwards correlated these information’s with the previous patients medical record. Therefore we consider this study and it’s design as a retrospective cohort observational study.

Results:

Lines 21-27: Information’s concerning the percentage of symptomatic patients with CBF changes were added. The paragraph has been reformatted to:

Among asymptomatic (A) and symptomatic (S) patients with carotid stenosis 3 subgroups were identified: 57/154 – 37% (A) and 8/31 – 25,5% (S) - with significantly increased flow volume (CBF higher than reference range: average CBF+std.dev in the group of healthy volunteers), 67/154 – 43,5% (A) and 12/31-39% (S) - with similar to reference group flow volume (CBF within range average±std.dev), and 30/154 – 19,5% (A) and 11/31-35,5% (S) - with decreased flow volume in ex-tracranial arteries (flow lower than average-std.dev. in healthy volunteers).

Lines 32-33: information’s about the number and percentage of asymptomatic and symptomatic patients were added:

…with similar degree of the ICA stenosis (8/31 – 25,8 % vs 26/53 – 49%, p=0,04).

Introduction:

The introduction was rewritten: the paragraphs concerning aim of the study was moved to the end of the introduction and merged with the last paragraph. Please, find the amended form below:

Lines 67-75: The aim of this study was to propose a novel approach of stratifying risk of neuro-logical incidents: a global cerebral inflow assessment in Doppler Ultrasonography (DUS), which may change current policy and indications for surgical intervention. It is based on characterisation of the flow volume in extracranial vessels: ICA, ECA and VA in patients >65 years old, with ≥50% carotid stenosis. Additionally, to prove the clinical significance of compensatory mechanisms, we compared the flow between asymptomatic patients and the symptomatic ones, with ICA stenosis ≥70%, who were referred for surgical treatment in order to assess whether the flow volume changes correlates with the incidence of neurological symptoms.

Materials and Methods:

The table with reference values of blood flow volume in extracranial arteries as well as its description have been moved to the results – a new section have been added. 

Line 150: 3.1 CBF reference values

Lines 111-116: Information about the methodology have been added:

Firstly, the diameter of each vessel was measured using three different techniques: B-mode, SMI (superb micro-vascular imaging) mode and B-mode combined with SMI image. The average of three measurements was considered a diameter of the vessel. Con-secutively the flow volume in the artery was measured with the ultrasound scanner sem-iautomatic program (the flow volume was calculated based on the spectral doppler and the diameter of the vessel) – see figure 1.

Figure 1 presenting the method of CBF measurement has been added

Figure 2 presenting an example of 67 years old patient with severe >95% ICA stenosis, and without flow compensation in other extracranial arteries, resulting with diminished CBF, have been added.

Results:

The following subsections were added:

Line 151: 3.1 CBF reference values.

Line 192: 3.2.1. CBF in the group with 50-69% ICA stenosis.

Line 196: 3.2.2. CBF in the group with 70-99% ICA stenosis. 

Line 206: 3.2.4. CBF in merged group ICA 70-99% stenosis + ICA occlusion.

In this paragraph the figure 3 was added. It presents graphically the changes in the percentage of patients with different CBF compensation level.

Line 216: 3.3. Comparison of the flow volume values and flow changes in asymptomatic patients ≥50% ICA stenosis.

Lines 228: 3.4. Pathways of volumetric flow compensation in the group of asymptomatic patients with ≥50% ICA stenosis.

Discussion:

The discussion was rewritten. The paragraphs summarizing our results and findings were moved at the beginning of the discussion.

Lines 287-318: The main finding of our study is identifying that changes in CBF correlate with the incidence of ischemic symptoms in patients with ICA stenosis.

Among patients with carotid artery stenosis, both: asymptomatic and symptomatic the three subgroups with volumetric flow changes were identified: patients with elevated CBF as a result of multivessel significant flow increase, patients with mild, less pronounced, compensation in whom the CBF is within proposed reference standard for healthy population, and patients without compensation, with CBF lower than healthy, equally aged population.

The percentage patients with flow compensation tend to increase with the severity of stenosis, and slightly decreases in the occlusion group. At the same time the percentage of patients without flow compensation does not significantly change and remain relatively constant, between 18,9% and 20%. The increase in percentage of people with compensatory elevated flow is accompanied by simultaneous decrease of patients with flow volume values “within proposed standard”.

What is prominent, in the group referred for surgical treatment (symptomatic, ≥70% ICA stenosis) the percentage of patients with flow compensation is twice as low as in the asymptomatic ones with similar degree of the ICA stenosis.

Aiming to highlight the idea of our study, the flow volumes between all age sub-groups in patients with ICA stenosis equal to or more than 70% were compared. It shows that asymptomatic patients tend to have higher CBF than symptomatic groups. The shortcoming of this comparison is relatively small number of patients (the study design demands separate assessment of patients in different age groups 65-69, 70-74, 75-79 and ≥80 years old), therefore, despite statistically significant differences and clearly identifiable tendency we do not want to draw any conclusion concerning numbers and amount of flow compensation. Further studies on much larger groups of patients are required.

In our study the decreased flow volume in extracranial arteries, the decreased CBF, is more frequently observed among symptomatic patients. While the tendency is clearly visible, the analysis of absolute numbers is very close to statistical significance, but due to relatively small number of patients does not reach it in chi2 test (p = 0,06 and 0,08). If both groups would be 2 times larger, than p values would be below 0,05. It may indicate that asymptomatic patients with lower values of CBF – without volumetric flow compensation, may be featured with higher risk of occurrence of ischemic symptoms.

Lines 413-420: a paragraphs concerning the three dimensional ultrasound have been added together with 4 new references [32-35]:

A novel tool in the diagnostic of carotid artery disease is a three dimensional ultra-sound. Contemporarily it is mainly used for carotid plaque assessment: its morphology, the presence of ulceration, which are correlated with risk of developing ischemic symp-toms. It is featured with high interobserver and interobserver reproducibility and can identify plaque volume changes as low as 4%–6% with 95% confidence [32-34].

It has been recently proven that three dimensional ultrasound may be a novel, prom-ising, potentially easily accessible and accurate tool in quantification of volumetric blood flow[35].

Lines 426-430: A summarizing paragraph was added:

Our method may allow to identify asymptomatic patients, with significant ICA ste-nosis and diminished CBF. Such patients may be more prone to developing ischemic and perhaps may benefit from surgical intervention. However, such conclusions cannot be drawn based on current work. A prospective study on a larger groups of patients should be conducted to investigate this issue.

References:

Lines 511-521: 4 new references were added

  1. Calogero E, Fabiani I, Pugliese NR, Santini V, Ghiadoni L, Di Stefano R, Galetta F, Sartucci F, Penno G, Berchiolli R, Ferrari M, Cioni D, Napoli V, De Caterina R, Di Bello V, Caramella D. Three-Dimensional Echographic Evaluation of Carotid Artery Disease. J Cardiovasc Echogr. 2018 Oct-Dec;28(4):218-227.
  2. Makris GC, Lavida A, Griffin M, Geroulakos G, Nicolaides AN. Three-dimensional ultrasound imaging for the evaluation of carotid atherosclerosis. 2011;219:377–83
  3. Hossain MM, AlMuhanna K, Zhao L, Lal BK, Sikdar S. Semiautomatic segmentation of atherosclerotic carotid artery wall volume using 3D ultrasound imaging. Med Phys2015;42:2029–43.
  4. Kripfgans OD, Pinter SZ, Baiu C, Bruce MF, Carson PL, Chen S, Erpelding TN, Gao J, Lockhart ME, Milkowski A, Obuchowski N, Robbin ML, Rubin JM, Zagzebski JA, Fowlkes JB. Three-dimensional US for Quantification of Volumetric Blood Flow: Multisite Multisystem Results from within the Quantitative Imaging Biomarkers Alliance. Radiology. 2020 Sep;296(3):662-670.

I hope You will find the revised version of manuscript suitable for publication in the Journal of Clinical Medicine.

Faithfully Yours,

Piotr Kaszczewski

Round 2

Reviewer 1 Report

The Authors addressed all the issues.

Author Response

Dir Sir or Madam,

                On behalf of all authors I would like to thank you for Your time and effort devoted to prepare the review of our work. I would also like to thank you for the suggestion how to improve our work. We put all our efforts to follow include them in our manuscript. 

                Following the suggestions of the Reviewer 2 we have implemented several amendments to our manuscript.     

The detailed list of the changes in the manuscript include:

  1. The description of the table 3 was changed to: “The differences in groups without flow compensation do not reach any significant level. The detailed data are presented in Table 3.”
  2. Figure 1 has been rearranged – the figure 1E presents the flow volume values in all study groups.
  3. The table 5 with p values observed in the figure 1 has been added to the manuscript.

Figure 1. CBF volume values in the asymptomatic patients with internal carotid ar-tery stenosis and occlusion (merged subgroups with different degree of ICA stenosis and occlusion). A: 65-69 years old, B: 70-74 years old, C: 75-79 years old, D: ≥80 years old, E – whole group merged together. P values are presented in the table 5.

  1. Paragraph 3.3 with figure 2 and table 7 has been added to the manuscript:

In the subgroup with ICA stenosis ≥70% statistically lower flow volume was observed in symptomatic patients in comparison with asymptomatic ones.

The data concerning the differences in the flow volume between symptomatic and asymptomatic patients with ≥70% ICA stenosis are presented in the Figure 2 and Table 7.

The group aged 75-79 was excluded from the analysis (in the symptomatic group there were only 2 patients).

In age groups 65-69, 70-74 and >80 years old statistically significant differences in flow volume were observed.

Figure 2. The differences in the flow volume between symptomatic and asymptomatic pa-tients with ≥70% ICA stenosis. A – age group 65-69 years old. B – age group 70-74 years old. C – age groups ≥80 years old.

Age group

Clinical presentation

Mean [ml/min]

25%-75%

[ml/min]

Min – Max

[ml/min]

p value

65-69

Asymptomatic

902,5

833,5-965,5

761-1172

p=0,049

Symptomatic

777,5

732-930,5

656-1201

70-74

Asymptomatic

884

756,5-981

679-1298

p=0,041

Symptomatic

763

692-878

530-1057

≥80

Asymptomatic

891,5

638-1092

571-1129

p=0,044

Symptomatic

607

587-709,5

570-809

Table 7. Detailed data concerning differences in the flow volume between symptomatic and asymptomatic patients with ≥70% ICA stenosis

The p values was added to the results.

Comparing asymptomatic and symptomatic patients with stenosis of 70% – 99% a statistically significant difference (p=0,0413) in patients with flow compensation is observed: 26/53 – 49% vs 8/31 – 25,8 % in those referred for surgical treatment. The relative risk of observing the compensatory increased flow in extracranial arteries in asymptomatic patients (n=53) is almost 2 times higher than in symptomatic group (n=31) - RR = 1,9; p<0,0413. Patients without flow compensation constituted 35,5% of referred for interventional procedure and only 19,3% (p=0,08) of asymptomatic patients with more than 70% stenosis and occlusion group, and 18.9 % (p=0,06) of patients with ICA stenosis 70-99%.

  1. The following paragraph discussing newly obtained results has been added to the discussion part:

Aiming to highlight the idea of our study, the flow volumes between all age sub-groups in patients with ICA stenosis equal to or more than 70% were compared. It shows that asymptomatic patients tend to have higher CBF than symptomatic groups. The shortcoming of this comparison is relatively small number of patients (the study design demands separate assessment of patients in different age groups 65-69, 70-74, 75-79 and ≥80 years old), therefore, despite statistically significant differences and clearly identifiable tendency we do not want to draw any conclusion concerning numbers and amount of flow compensation. Further studies on much larger groups of patients are required.

In our study the decreased flow volume in extracranial arteries, the decreased CBF, is more frequently observed among symptomatic patients. While the tendency is clearly visible, the analysis of absolute numbers is very close to statistical significance, but due to relatively small number of patients does not reach it in chi2 test (p = 0,06 and 0,08). If both groups would be 2 times larger, than p values would be below 0,01. It may indicate that asymptomatic patients with lower values of CBF – without volumetric flow com-pensation, may be featured with higher risk of occurrence of ischemic symptoms. Our method may allow to identify such patients, who perhaps may benefit from surgical intervention, however such conclusions cannot be drawn based on current work. A prospective study on a larger groups of patients should be conducted to investigate this issue.

  1. The conclusion 2 has been slightly modified:

The assessment of global cerebral inflow in Doppler Ultrasonography may provide novel and easily accessible tool of identifying patients more prone to cerebral ischaemia – further studies on a larger groups of patients are required.

We do hope that you will consider the amended version of the manuscript suitable for publication in the Journal of Clinical Medicine.

Faithfully Yours,

Piotr Kaszczewski

Michał Elwertowski

Jerzy Leszczyński

Tomasz Ostrowski

Zbigniew Gałązka

Reviewer 2 Report

Dear authors,

I'm fully aware of your good intention for your patients. I'm also a physician who do research as well. However, the good intention is NOT justified enough for a paper to be published. For example, how would the readers appreciate the value of "spared many of our patients from stroke and death – we feel clinical impact of this work is the most important factor that should be brought to everybody’s attention", based on the currently, most descriptive study? The study results are cross-sectional, with merely observed association without causal relationship. Not to say there are significant overlap of the proportion of "no compensation", "mild compensation" in either asymptomatic or symptomatic groups. Therefore readers may not be able to realize when facing an asymptomatic patient with >70% stenosis but "mild compensation" of flow, what should they do then? Refer for surgical or keep medical treatment? The study may not be able to answer. Anyway, the intention of the study is NOT justified with its result.

Still several points need to be improved:

1) The authors mentioned about statistical analyses, such as Mann-Whiteny U test, Kruskal-Wallis test... However no any p value was presented in the table. Only descriptions such as "substantial differences", "a statistical significant difference" were presented. The authors may read some other clinical papers to see how to present the group differences in the table.

2) The Figure 1 did not provide additional information. In each age group, the flow difference between "reference", "mild compensation", "significant compensation" and "no compensation" are inherently different since you already GROUP them in such a way! The rationale of age stratification is not described either. I assumed the authors tried to explore whether the compensation would be different in each age group. In doing so, the authors should plot all Figure 1A-D together, to see whether the compensation flow in the significant compensation groups were quantitative different in each age group. Otherwise, the readers still cannot get much information from the Figure.

3) The authors stated that the baseline comparison was done in their previous paper. It's not a proper excuse for NOT mentioning again in this paper. See, even in the subgroup analysis paper of a major clinical trials (for example, the follow-up subgroup analysis of a MR CLEAN trial), they will still present the baseline characteristics of two groups, in order to let the readers to rapidly catch up the idea. Simply providing those in the table won't make your manuscript lengthy, but rather complete it. Besides, providing vascular risk factors allow the readers (and the authors too) to scrutinize whether those factors are different between groups, and a multivariable regression may then be applied.

4) Regarding the regression, the authors' reply that it may require a significantly larger sample size doesn't make sense either. Your study already recruited 154 asymptomatic patients and 31 symptomatic patients. Those numbers are quite good enough for multivariable regression analysis. For example, the canonical "one in ten rule" implied that you may use up to 18 variables with 180 patients. I don't quite get the idea of "We would like to present our results in one single work, however the very “Results“ paragraph, will have about 15000 words, and in such work the main, clinical idea will be lost among mathematics, statistics, numbers and figures." In anyway, wasn't that the training of clinical scientific paper requires the authors to extract and concentrate on the most important findings from their extensive work? 

5) If your take-home idea (the last sentence in the conclusion) is "The multivessel character of compensation with enhanced role of ECA stresses the 338 importance of including this artery in the estimation of CBF", then it is NOT justified of not further analyzing ECA flow. If that's in the context of another project, then it's better not to overemphasize in the current one. However, it is interesting that since the CBF in the current paper was based on the grand total of the flow volume of ICA, ECA and VA, the authors should be able to provide the detail numbers of those 3 vessels in each group.

After considering of your study aim again, you may consider reanalyze your data in the following ways to improve the soundness of the study:

1) Compare the baseline characteristics and their compensatory flow status between the four groups: asymptomatic groups*3 (50-69%, 70-99%, 100% occlusion) and the symptomatic group, using chi-square test and Kruskal-Wallis test. 

2) Direct compare those with asymptomatic and symptomatic >70% stenosis (n=53 and 31 respectively?), to see if the compensatory flow are equivalent or not between the two groups. (Your current analysis is kind of this way, but you only use "flow compensation", but not the absolute number of flow amount. You didn't even provide the P value.) And may use regression model to see what factors (eg., vascular risk factors, flow amount, compensatory flow vessel...) are associated with symptomatic stenosis. 

Anyway, I truly appreciate the intention of the author group. However to be published in a prestigious journal, significant improvement is still required in my opinion. Thank you again.

Author Response

Dir Sir or Madam,

                On behalf of all authors I would like to thank you for Your time and effort devoted to prepare the review of our work. I would also like to thank you for the suggestion how to improve our work. We put all our efforts to follow include them in our manuscript.  

                At the beginning of your second review you asked a question what to do with patient with mild compensation? Nowadays the patients referred for surgery should have symptomatic >70% ICA stenosis. Our research aims to show that in asymptomatic patients with >70% ICA stenosis, who have volumetric flow compensation, and their flow is higher than in healthy equally aged population, the risk of occurrence of ischaemic symptoms is lower. That is one of our conclusions. Significantly lower flow volume values in symptomatic groups allow to presume that, perhaps patients with lower flow volume values should be treated earlier, however it deserves randomized clinical study.

                As you suggested we have prepared the combined plot in the figure one together with table presenting p values between groups. We didn’t want to analyse the degree of flow compensation, we wanted to show that within every age group with ICA stenosis, there are patients with flow lower than, equal to or higher than the one in healthy equally population.

                The rationale of such data presentation and division of the study group in cohorts in 5 years intervals comes from our previous study. We cannot analyse the data in a different way if our reference values are established for such age groups. The baseline reference group characteristics is presented in the table 1.

                Dir Sir, I would like to explain why we didn’t do the baseline characteristics of the groups as you asked:

“Compare the baseline characteristics and their compensatory flow status between the four groups: asymptomatic groups*3 (50-69%, 70-99%, 100% occlusion) and the symptomatic group, using chi-square test and Kruskal-Wallis test”.

What you proposed is a great idea, that could beyond any doubts prove the clinical significance of the study. However, to perform it, we should do separate analysis for each age group: 4 groups (because as we showed in the previous study, flow volume changes significantly with increasing age). To present the compensatory status we should divide them further into three subgroups based on compensation status (4x additional 3 groups = 12 groups), and further based on stenosis degree: another (12 x 3 groups = 36 groups). We have already 36 subgroups of asymptomatic patients. We have only 31 symptomatic patients. If we divide this group even into 3 age groups – we compare the groups of about 10 patients. We cannot draw any conclusions basing on such small study groups analysis. Our number of patients does not allow for subgroup analysis. But as for ultrasound study our study group is one of the largest in medical literature.

                Frankly speaking we faced the same dilemmas preparing the figure2, which is answer to your second request. The p values are significant, the tendency is in accordance with our study and observations, but we compare relatively small groups of patients, and it is something we didn’t want to do. Therefore, we added a paragraph to our discussion describing that, despite this analysis really highlights what we wanted to prove, and clearly shows flow differences, it should be performed on a much larger groups of patients. We are not able to draw representative conclusions based on this analysis.

We didn’t want to analyse symptomatic patients in this study. We wanted to compare the percentage of asymptomatic patients with flow compensation with the percentage of symptomatic patients with flow compensation, to prove the clinical aspect of the research. Our conclusion is:  “The significant compensatory elevation of CBF was more frequently observed in asymptomatic patients, which suggest its protective influence on developing ischaemic symptoms”.  If we analysed only asymptomatic patients our study would make no sense. Asymptomatic patients in overwhelming majority should be treated conservatively with BMT (unless they have microembolisation, rapid stenosis progression etc.). With our method we may identify patients with lower CBF, in whom ischaemic symptoms are more frequent. Whether to treat them or not requires further investigation. Our experience and observations allows us to claim that such direction of investigation may have clinical sense and importance.

The detailed list of the changes in the manuscript include:

  1. The description of the table 3 was changed to: “The differences in groups without flow compensation do not reach any significant level. The detailed data are presented in Table 3.”
  2. Figure 1 has been rearranged – the figure 1E presents the flow volume values in all study groups.
  3. The table 5 with p values observed in the figure 1 has been added to the manuscript.

Figure 1. CBF volume values in the asymptomatic patients with internal carotid ar-tery stenosis and occlusion (merged subgroups with different degree of ICA stenosis and occlusion). A: 65-69 years old, B: 70-74 years old, C: 75-79 years old, D: ≥80 years old, E – whole group merged together. P values are presented in the table 5.

  1. Paragraph 3.3 with figure 2 and table 7 has been added to the manuscript:

In the subgroup with ICA stenosis ≥70% statistically lower flow volume was observed in symptomatic patients in comparison with asymptomatic ones.

The data concerning the differences in the flow volume between symptomatic and asymptomatic patients with ≥70% ICA stenosis are presented in the Figure 2 and Table 7.

The group aged 75-79 was excluded from the analysis (in the symptomatic group there were only 2 patients).

In age groups 65-69, 70-74 and >80 years old statistically significant differences in flow volume were observed.

Figure 2. The differences in the flow volume between symptomatic and asymptomatic pa-tients with ≥70% ICA stenosis. A – age group 65-69 years old. B – age group 70-74 years old. C – age groups ≥80 years old.

Age group

Clinical presentation

Mean [ml/min]

25%-75%

[ml/min]

Min – Max

[ml/min]

p value

65-69

Asymptomatic

902,5

833,5-965,5

761-1172

p=0,049

Symptomatic

777,5

732-930,5

656-1201

70-74

Asymptomatic

884

756,5-981

679-1298

p=0,041

Symptomatic

763

692-878

530-1057

≥80

Asymptomatic

891,5

638-1092

571-1129

p=0,044

Symptomatic

607

587-709,5

570-809

Table 7. Detailed data concerning differences in the flow volume between symptomatic and asymptomatic patients with ≥70% ICA stenosis

The p values was added to the results.

Comparing asymptomatic and symptomatic patients with stenosis of 70% – 99% a statistically significant difference (p=0,0413) in patients with flow compensation is observed: 26/53 – 49% vs 8/31 – 25,8 % in those referred for surgical treatment. The relative risk of observing the compensatory increased flow in extracranial arteries in asymptomatic patients (n=53) is almost 2 times higher than in symptomatic group (n=31) - RR = 1,9; p<0,0413. Patients without flow compensation constituted 35,5% of referred for interventional procedure and only 19,3% (p=0,08) of asymptomatic patients with more than 70% stenosis and occlusion group, and 18.9 % (p=0,06) of patients with ICA stenosis 70-99%.

  1. The following paragraph discussing newly obtained results has been added to the discussion part:

Aiming to highlight the idea of our study, the flow volumes between all age sub-groups in patients with ICA stenosis equal to or more than 70% were compared. It shows that asymptomatic patients tend to have higher CBF than symptomatic groups. The shortcoming of this comparison is relatively small number of patients (the study design demands separate assessment of patients in different age groups 65-69, 70-74, 75-79 and ≥80 years old), therefore, despite statistically significant differences and clearly identifiable tendency we do not want to draw any conclusion concerning numbers and amount of flow compensation. Further studies on much larger groups of patients are required.

In our study the decreased flow volume in extracranial arteries, the decreased CBF, is more frequently observed among symptomatic patients. While the tendency is clearly visible, the analysis of absolute numbers is very close to statistical significance, but due to relatively small number of patients does not reach it in chi2 test (p = 0,06 and 0,08). If both groups would be 2 times larger, than p values would be below 0,01. It may indicate that asymptomatic patients with lower values of CBF – without volumetric flow com-pensation, may be featured with higher risk of occurrence of ischemic symptoms. Our method may allow to identify such patients, who perhaps may benefit from surgical intervention, however such conclusions cannot be drawn based on current work. A prospective study on a larger groups of patients should be conducted to investigate this issue.

  1. The conclusion 2 has been slightly modified:

The assessment of global cerebral inflow in Doppler Ultrasonography may provide novel and easily accessible tool of identifying patients more prone to cerebral ischaemia – further studies on a larger groups of patients are required.

We do hope that you will consider the amended version of the manuscript suitable for publication in the Journal of Clinical Medicine.

Faithfully Yours,

Piotr Kaszczewski

Michał Elwertowski

Jerzy Leszczyński

Tomasz Ostrowski

Zbigniew Gałązka
